# Analysis of the Depth of Immersion of the Submerged Entry Nozzle on the Oscillations of the Meniscus in a Continuous Casting Mold

**F. Saldaña-Salas [1], E. Torres-Alonso [2], J.A. Ramos-Banderas [2,*], G. Solorio-Díaz [1] and C.A. Hernández-Bocanegra [2,3]**

[1]  UMSNH, Posgrado en Ingeniería Mecánica, Av. F.J. Mujica S/N, Morelia 58040, Mexico;
    fernando_4_6@hotmail.com (F.S.-S.); gil_6000@hotmail.com (G.S.-D.)
[2]  TecNM/I.T. Morelia, Av. Tecnológico #1500, Col. Lomas de Santiaguito, Morelia 58120, Mexico;
    enriquetorresalonso@hotmail.com (E.T.-A.); beto.constan@gmail.com (C.A.H.-B.)
[3]  CATEDRAS-CONACyT, Av. Insurgentes Sur # 1582, CDMX 03940, Mexico
*   Correspondence: arblss@hotmail.com or jaramos@itmorelia.edu.mx; Tel.: +52-443-3121570 (ext. 300)

**Abstract:** In this study the effects of the depth of immersion of the Submerged Entry Nozzles (SEN) on the fluid-dynamic structure, oscillations of the free surface and opening of the slag layer, in a continuous casting mold for conventional slab of steel were analyzed. For this work, a water/oil/air system was used in a 1:1 scale model, using the techniques of Particle Image Velocimetry (PIV), colorimetry and mathematical multiphase simulation. The results of the fluid dynamics by PIV agree with those obtained in the mathematical simulation, as well as with the dispersion of dye. It was observed that working with immersion depths of 100 mm or less could be detrimental to steel quality because they promote surface oscillations of a higher degree of Stokes with high elevations and asymmetry in their three dimensions. In addition, this generates an excessive opening of the oil layer which was corroborated through the quantification of the F index. On the other hand, with depths of immersion in the range of 150–200 mm, lower oscillations were obtained as well as zones of low speed near the wall of the SEN and a smaller opening of the oil layer.

**Keywords:** continuous casting mold; PIV; mathematical simulation; immersion of the SEN; level oscillations

---

## 1. Introduction

The production and consumption of steel has increased extraordinarily in the last decades worldwide, occupying the first place the Asian continent with ~68% of the production [1]. This demand justifies the extensive research of the continuous casting process to increase steel quality and/or productivity levels. This process was patented in the 1950s [2] and more than 95% of the steel is produced by this route. In general, it consists of taking the molten steel from the Basic Oxygen Furnace (BOF) or the Electric Arc Furnace (EAF) to the secondary refining station, where chemical composition and temperature adjustments are made. Once it reaches customer specifications, it is poured from the ladle to the tundish, allowing refining operations. Finally, the steel flows from the tundish to the mold through a sliding valve and a Submerged Entry Nozzle (SEN) to finally solidify continuously in the form of slab, billet or bloom in a water-cooled mold, for its consequent transformation process [3].

During the manufacture of the steel, an intense erosion of the outer wall of the SEN has been detected, mainly the one that comes in contact with the fluxes that form the slag of the mold [4]. The mechanisms of wear have already been described [5] and discussed in later studies [6–10]. The wear of the SEN directly impacts the productivity of the plant [11] and has been tried to be

eliminated by adapting the chemical composition of the slag according to the type of steel that is produced [12]. On the other hand, antioxidant coatings are also used in the area of contact with slag such as SiC or silicon metal [13]. $ZrO_2$ stabilized with MgO, $Y_2O_3$ or CaO that promotes high resistance to corrosion and is used to minimize thermal shock. However, a study [8] concluded that antioxidants can destabilize $ZrO_2$, accelerating the process of erosion of the SEN. In this sense, it is important to mention that in industrial practice, the depth of immersion of the SEN is commonly varied to reduce the localized wear that occurs through contact with the slag line. Although, this practice allows to increase the useful life of the SEN and consequently the productivity of the plant, the surface quality of the steel slab can be compromised if there is no control of the immersion of the SEN.

However, there are reports [14] where the immersion of the SEN has been fixed, as well as the casting speed and even so, phenomena such as the periodic oscillation of the jets at the exit of the SEN ports or transient phenomena are observed. Dynamic distortion phenomena related to the dissipation of turbulent kinetic energy during the casting of thin slab has also been reported [15]. Therefore, the practice of varying the immersion of the SEN in the mold can result in large fluctuations in the surface of the steel, causing dragging of cover dust and impurities that can be trapped in the solidified shell, as well as promoting a non-uniform solidification [3,16–21]. Additionally, the thinning of the solidified skin that, in combination with a high superheat, could cause a line breakout [22]. Perhaps one of the most damaging defects reported in the slab due to the large variations in level in the mold are the longitudinal corner cracks, for which is recommended not to exceed 3 mm in the variation of level to avoid the appearance of this type of surface defect [23–25]. For these situations with respect to the stabilization of the level in the mold, some works have already been published that involve the design of level control systems [26–28], re-design of the lower supply ports of the SEN to minimize the perturbations of the level in the mold [29], which have been classified as periodic and non-periodic waves according to their origin [30].

To establish the limits on the permissible oscillation range of the steel level in the mold without affecting the surface quality of the slab, the F index has been developed; which is a level fluctuation factor in the meniscus area, which relates the momentum of the steel flow that rises to the surface once it impinges with the narrow wall of the mold [31,32].

Due to the importance of the fluid dynamics inside the mold and their level of the steel, when the immersion depth of the SEN is varied, an analysis of this phenomenon was carried out in this work. Physical modeling tools (colorimetry and Particle Image Velocimetry) and multiphase three-dimensional mathematical simulation in a transient state in a full-scale water/oil/air model were employed. Three immersion depths (100, 150 and 200 mm) of a bifurcated SEN were studied, considering the typical variations in the plant with the intention of avoiding the localized wear of the nozzle in contact with the slag, and the F index was quantified to associate it with the level oscillations in all cases of study.

## 2. Experimental

### 2.1. Mold Geometry

Figure 1a shows the dimensions of the calculation domain and the details of the SEN by means of a front view. Figure 1b shows the planes $P_{III}$ and $P_V$ used for the measurement of the fluid dynamic, and the planes $P_I$, $P_{II}$, $P_{III}$ and $P_{IV}$ are shown to analyze the level oscillations. The $P_I$ and $P_{IV}$ planes are located at the fluid-wall interface.

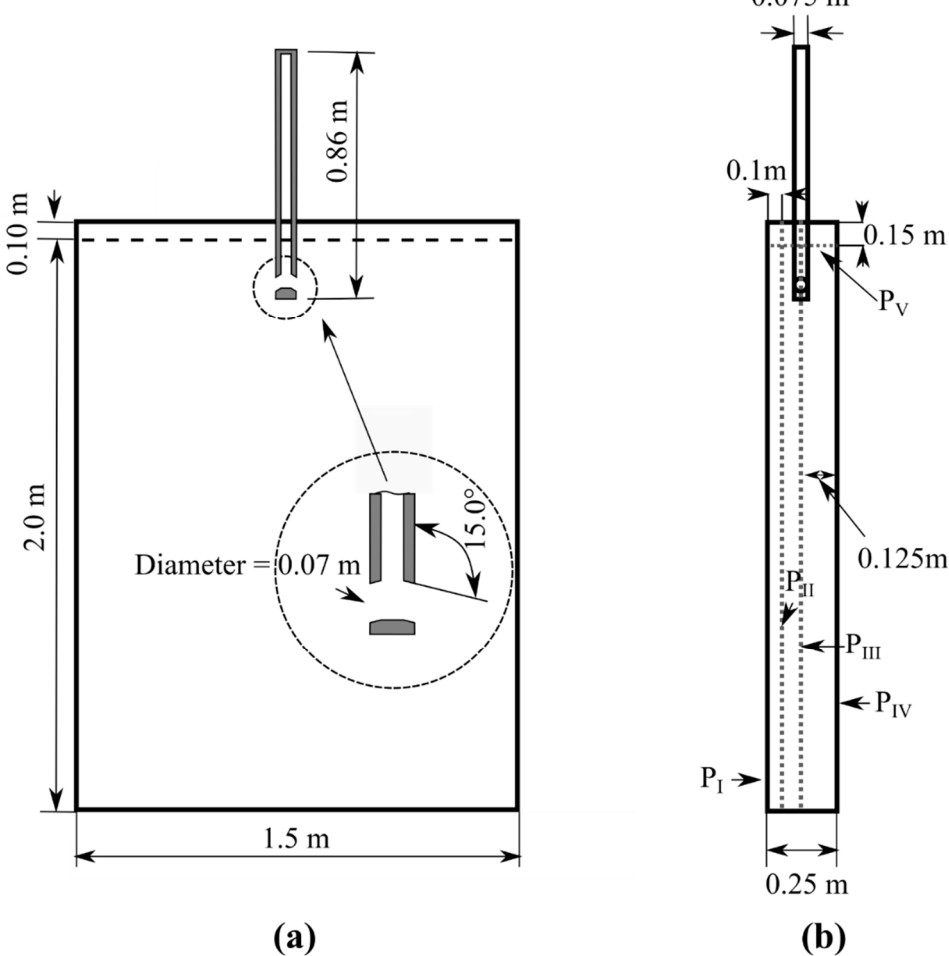

**Figure 1.** Geometrical mold and Submerged Entry Nozzle (SEN) dimensions and planes of analysis; (**a**) frontal view and (**b**) lateral view.

## 2.2. Mathematical Formulation and Assumptions

The flow field in the mold was calculated by solving the conservation mass and momentum equation in a 3D coordinate system, adjusting the control volume to a set of boundary conditions. The considerations were as follows: an incompressible and Newtonian fluid, completely turbulent flow inside the mold, transient state and for the case of solid walls, a non-slip condition was used. To solve the fluid dynamics of the system, a set of Navier-Stokes equations adapted to multiphase flow using Volume of Fluid (VOF) model was employed.

### 2.2.1. Continuity Equation

The VOF model can simulate two or more immiscible fluids by solving a single set of momentum equations and tracking the volume fraction of each of the fluids throughout the domain. A continuity equation has to be solved for the volume fraction of one or more of the phases [33].

$$\frac{\partial}{\partial t}\left(\alpha_q \rho_q\right) + \nabla \cdot \left(\alpha_q \rho_q \vec{u}_q\right) = 0, \tag{1}$$

where subscript $q$, stands for the different phases (air, water, vegetable oil); the value of $\alpha$ defines the phase state of a control cell in the computational mesh:

$$\sum_{q=1}^{n} \alpha_q = 1. \tag{2}$$

### 2.2.2. Momentum Equation

A single momentum equation is solved throughout the domain, and the resulting velocity field is shared among the phases. The momentum equation, shown below, is dependent on the volume fractions of all phases through the properties $\rho$ and $\mu$.

$$\frac{\partial}{\partial t}\left(\rho \vec{u}\right) + \nabla \cdot \left(\rho \vec{u}\vec{u}\right) = -\nabla p + \nabla \cdot \left[\mu\left(\nabla\vec{u} + \nabla\vec{u}^T\right)\right] + \rho\vec{g}, \tag{3}$$

where $\rho$ is the fluid density in kg·m$^{-3}$; $\vec{u}$ represents the velocity in m·s$^{-1}$; $t$ is the time, $p$ and g represent the pressure in Pa and gravity acceleration (9.81 m·s$^{-2}$), respectively; and $\mu$ is the viscosity in Pa·s.

### 2.2.3. Turbulence Equations

The Reynolds Stress Model (RSM) [34], was used to solve turbulence conditions which are indicated in Equation (4). For better understanding, all variables and Greek letters have been described at Table A1.

$$\underbrace{\frac{\partial}{\partial t}\left(\rho\overline{u_i u_j}\right)}_{\text{Local Time Derivative}} + \underbrace{\frac{\partial}{\partial x_k}\left(\rho u_k \overline{u_i u_j}\right)}_{C_{ij} \equiv \text{Convection}} = \underbrace{-\frac{\partial}{\partial x_k}\left[\rho\overline{u_i u_j u_k} + \overline{\rho\left(\varsigma_{kj}u_i + \varsigma_{ik}u_j\right)}\right]}_{DT,ij \equiv \text{Turbulent Diffusion}} + \underbrace{\frac{\partial}{\partial x_k}\left[\mu\frac{\partial}{\partial x_k}\left(\overline{u_i u_j}\right)\right]}_{D_{L,ij} \equiv \text{Molecular Diffusion}} -$$

$$\underbrace{\rho\left(\overline{u_i u_k}\frac{\partial u_j}{\partial x_k} + \overline{u_j u_k}\frac{\partial u_i}{\partial u_k}\right)}_{P_{ij} \equiv \text{Stress Production}} - \underbrace{\rho\beta\left(g_i\overline{u_j\theta} + g_j\overline{u_i\theta}\right)}_{G_{ij} \equiv \text{Bouyancy Production}} + \underbrace{\overline{p\left(\frac{\partial u_i}{\partial x_j} + \frac{\partial u_j}{\partial x_i}\right)}}_{\varphi_{ij} \equiv \text{Pressure Strain}} - \underbrace{2\mu\frac{\overline{\partial u_i}}{\partial x_k}\frac{\partial u_j}{\partial x_k}}_{\varepsilon_{ij} \equiv \text{Dissipation}} \tag{4}$$

$$\underbrace{-2\rho\Omega_k\left(\overline{u_j u_m}\varepsilon_{ikm} + \overline{u_i u_m}\varepsilon_{jkm}\right)}_{F_{ij} \equiv \text{Production by System Rotation}} + \underbrace{S_{user}}_{\text{User} - \text{Defined Source Term}}$$

### 2.3. Boundary Conditions

Figure 2 shows the boundary conditions, where the exit velocity was obtained through Equation (5) having a constant casting speed of 1 m·min$^{-1}$, it was initialized with a constant level of 1.9 m and a thickness of the oil layer of 0.015 m was simulated.

$$Q = V_{inlet} \cdot A_{inlet} = V_{casting} \cdot A_{outlet}. \tag{5}$$

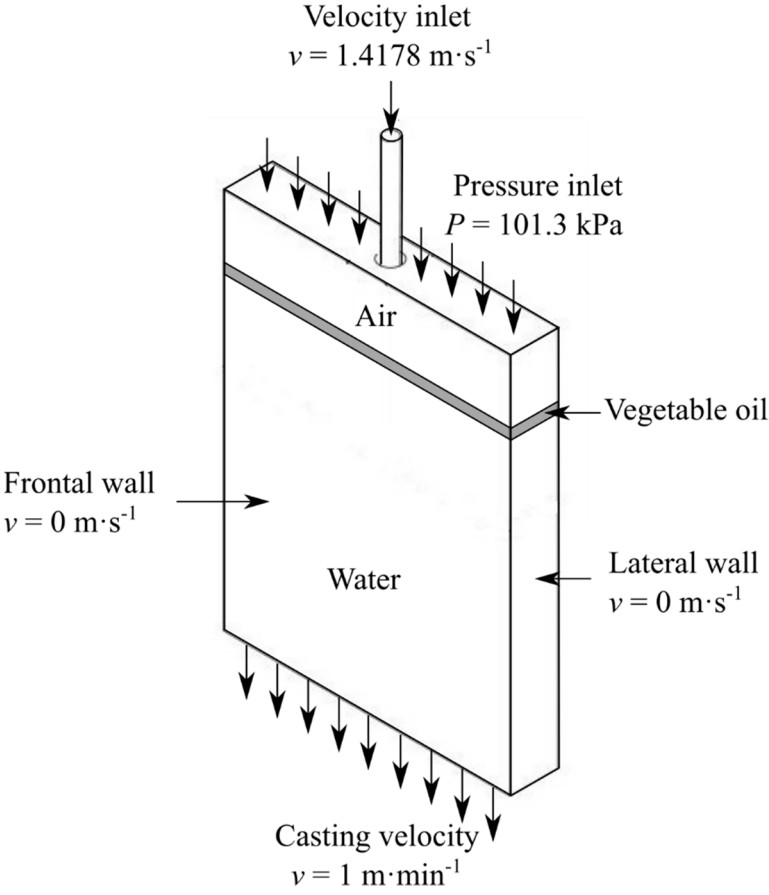

**Figure 2.** Initial and boundary conditions.

## 2.4. Processing

The computational domain was discretized in 1,044,468 structured elements. The simulated cases for the three immersion depths of the 100, 150 and 200 mm are referred to as Cases I, II and III, respectively; 8 min were simulated with a step size of 0.01 s, using the PISO pressure-velocity coupling algorithm [35]. Convergence was established when the residuals reached values below $1 \times 10^{-5}$. All equations were solved using ANSYS FLUENT® (version 15, ANSYS Inc., Canonsburg, PA, USA). The steel at 1600 °C was simulated using water at room temperature because both liquids have a similar kinematic viscosity, and thus the criteria of similarity required in the theory of dimensional analysis and similarity are met [36]. The slag layer was simulated using vegetable oil. Table 1 shows the physical properties of the phases.

**Table 1.** Physical properties of fluids.

| Fluid | Density ($\rho$) kg·m$^{-3}$ | Viscosity ($\mu$) kg ·m$^{-1}$·s$^{-1}$ |
|---|---|---|
| Water (20°) | 1000 | 0.001 |
| Vegetable Oil | 930 | 0.3534 |
| Air | 1.255 | $1.7897 \times 10^{-5}$ |
| **Interfacial Tension ($\sigma$) N·m$^{-1}$** | | |
| Water–air | | 0.073 |
| Water–Oil | | 0.035 |
| Oil–Air | | 0.033 |

## 3. Results and Discussion

### 3.1. Validation of the Model

To obtain the velocity fields in the scale model a Particle Image Velocimetry (PIV) technique was employed. A scheme of the PIV configuration (Dantec Systems ®, Skovlunde, Denmark) used in this study is shown in Figure 3. This equipment has a green laser Nd:YFL dual power 30–1000 with maximum pulse of $2 \times 30$ mJ and 20 Khz. To obtain pulses of light energy, the laser signal is interrupted and thus emit pulses of 250 μs, which is the duration of excitation of the lamp in the laser cavity. The power output of the laser is 150 watts and wavelength of 527 nm. The light sheet forming the laser was positioned in the plane indicated in Figure 1b by means of a positioner with three-dimensional movement controlled by the computer. The camera was positioned in a perpendicular form to the plane $P_{III}$ shown in Figure 1b. The fluorescent particles made of polyamide have a diameter of 50 μm with a density of 1030 kg·m$^{-3}$ and were incorporated into the fluid before initializing the system. A cross-linking procedure, using Fourier transforms, was used to process the recorded signals; and a Gaussian distribution function was used to locate the maximum peak displacement with an accuracy of the order of sub-pixels. The signals were recorded by a Speed Sense Phantom Miro M310 camera, which uses a Planar 1.4/50 mm ZF lenses, and then processed using the commercial software DynamicStudio provided by Dantec Systems (Dantec Dynamics, Skovlunde, Denmark), finally obtaining the velocity fields. The area of plane $P_{III}$ is $1500 \times 1000$ mm$^2$ and using a density of vector of $1280 \times 800$ pixels, 2000 images were acquired in 2 s. Finally, the images of the vector fields are the result of the averaged images. For the experimental validation, the same conditions of the mathematical model were used.

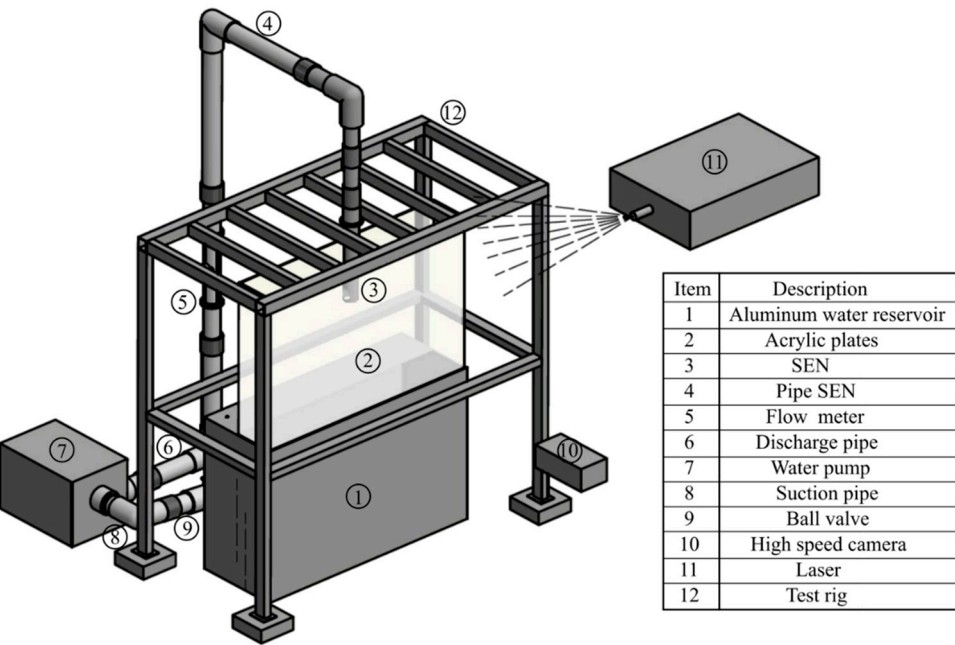

| Item | Description |
|------|-------------|
| 1 | Aluminum water reservoir |
| 2 | Acrylic plates |
| 3 | SEN |
| 4 | Pipe SEN |
| 5 | Flow meter |
| 6 | Discharge pipe |
| 7 | Water pump |
| 8 | Suction pipe |
| 9 | Ball valve |
| 10 | High speed camera |
| 11 | Laser |
| 12 | Test rig |

**Figure 3.** Experimental setup for Particle Image Velocimetry (PIV).

Figure 4a–c shows the flow dynamics inside the mold in the upper half of the $P_{III}$ plane, due to the restriction of the vision field of the PIV equipment. Figure 4a,b shows the velocity fields obtained by the $P_{IV}$ and mathematical simulation techniques, respectively. In Figure 4c the flow dynamics obtained through the colorimetry technique is shown. A great similarity in the fluid dynamic pattern by these techniques can be observed. In the three techniques, the jets coming out of the ports advance until they hit the narrow walls of the mold, which results in four large recirculations: two higher and two lower recirculations as indicated in the numbers "1, 2, 4 and 5," likewise there is an upward flow indicated by the number "3." In Figure 4a,b, it is observed that the higher velocities correspond to the

area of the jets, being the left one that impacts to a greater depth, and then an asymmetry in the jets is appreciated, which has already been reported for this type of SEN [14]. As can be seen, there is a great correspondence in terms of number, size, shape and location of the recirculations and ascending flow through the employed techniques.

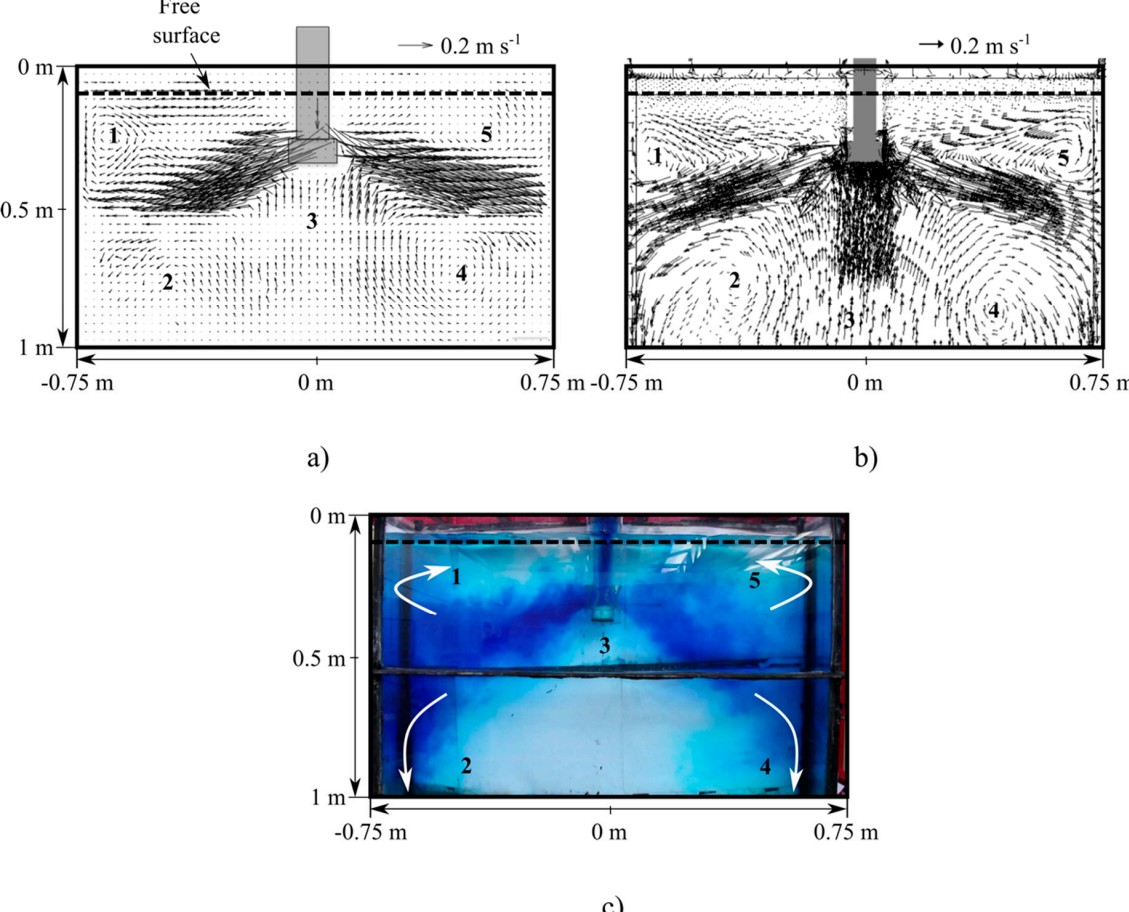

**Figure 4.** Velocity fields in the $P_{III}$ plane (**a**) PIV and (**b**) mathematical modelling. (**c**) Colorant dispersion technique.

A noteworthy consequence of the impact depth of the jet can be seen by comparing the higher recirculations with each other. That is, the higher speeds of the right side generate a greater recirculation near the free surface, with respect to the one formed on the left side, which agrees with Figure 4c (number 5).

## 3.2. Velocity Field

Once the mathematical model was validated, it was used to analyze the fluid dynamics structure of the three studied cases. Figure 5a–c shows the velocity fields in the $P_{III}$ plane for 410 s of simulation time. In general, the three cases present a typical predominant structure, which is already reported in the literature for bifurcated SEN, known as "double roll" represented by recirculations 1–4 for the three cases and an upward flow from the lower part of the mold [37]. However, there are some small variations to the "double roll" structure depending on the immersion depth of the SEN. In Cases I and II recirculation 5 appears; in Case I inside recirculation 4, and in Case II, independently below recirculation 3. On the other hand, in Case III, recirculation 5 disappears from the bottom of the mold but recirculations 5–8 are generated above the jets as shown in Figure 5c. Regarding the symmetry obtained in the fluid dynamic structure, very few differences are observed for Cases I and II, given the

occurrence of the new recirculations, Case III exhibit the best symmetry with respect to the central axis of the SEN. In relation to the velocity that is reached in the main upper recirculations, Case I exhibits the highest values, reaching the walls of the SEN at a very high velocity. Case III obtains the lowest velocity values in the free surface, which can be due to the appearance of its additional recirculations that dissipate the energy coming from the upward flow in this area. Additionally, very high values of velocity of the main lower recirculations are observed for Case I in relation to the other cases. After impacting the narrow wall, the jet remains with high velocity as it travels along this wall all the way to the bottom and goes up through the center to hit the lower wall of the SEN. On the contrary, as the SEN submerges (Cases II and III), the jet's velocity gradually slows down, once they hit the narrow wall, and the ascending flow accelerates as it interacts with the two lower main recirculations 3 and 4. This leads to the flow that feeds in the inferior corners of the mold, which is more noticeable in the Case III and that cushions the descending flow that advances along the narrow walls. On the other hand, in Figure 5a–c it is also observed that the structure developed for the outflow jets of the SEN for Case I is very thin, and becomes wider as the depth of immersion of the SEN increases (Cases II and III, respectively). The above is due to the fact that the high velocities of the main upper recirculations 1 and 2 for Case I impact a greater area of the jet structure through their upper part, and it is even possible to observe their deformation. This effect vanishes with increasing SEN immersion. For Case III the jet develops without alterations, losing kinetic energy uniformly by viscous dissipation as it approaches the narrow wall. As has been reported [32], the quality of steel—among other variables—depends on the solidification process and the flow feeding by the SEN. In other work [29] it was reported that the fluid dynamic structure depends exclusively on the design of the SEN, but due to the results found in this study it is possible to add that it also depends on the depth of immersion in the mold.

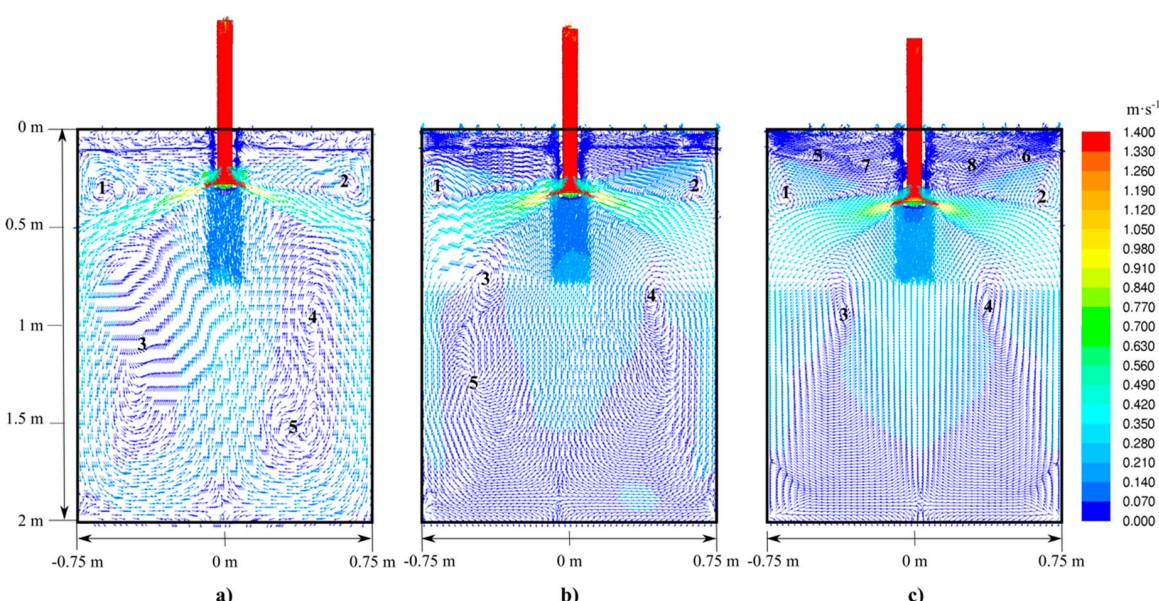

**Figure 5.** Velocity fields in the $P_{III}$ plane. (**a**) Case I, (**b**) Case II and (**c**) Case III.

Figure 6a–c shows the velocity field in $P_V$ plane for the three cases. Cases I and II show greater symmetry in the fluid dynamic pattern, compared to Case III, which is completely asymmetric. In turn, the velocities in this plane are greater than for the cases where the SEN is less submerged. In Case I, Figure 6a, it is observed how the fluid coming from the narrow faces of the mold advances toward the center of the mold at velocities of ~0.08 m·s$^{-1}$. There are also two recirculations, denoted by the numbers 1 and 2, very close to the wall of the SEN, that of the right side being larger and with higher velocities. The recirculation 2 is at the center while the recirculation 1 is located at the bottom respect to the wide walls, causing the fluid to cross the axis of symmetry, $L_S$. As a result of these, high velocities are generated around the SEN walls, which could diminish the refractory useful life. For Case II, the

fluid dynamics is completely different. In this case, it is observed how these two large recirculations disappear and six smaller ones are developed which are indicated in Figure 6b. It is observed that this case is the one that shows greater symmetry with respect to Ls, generating three recirculations on each side. The recirculations 1–4 are close to the SEN wall, causing a low speed zone in the central area of the studied plane. The above happens because the recirculations collide, causing a damping effect which decreases the intensity. It is also possible to observe at the ends of this plane that two large recirculations are formed that exhibit high velocity due to the fluid that ascends through the narrow walls. In Figure 6c, Case III, it can be seen that the low velocity zone is wider than the previous cases, occupying ~ 2/3 of the total area of the plane. In points 1 and 2 it is observed that the fluid enters this plane coming from the bottom with high velocities, redirecting towards the narrow walls and impacting on them with magnitudes ~0.03 m·s$^{-1}$. It has been reported [38] that velocities of ~0.2–0.3 m·s$^{-1}$ or greater allow the formation of vortices that can drag the slag into the metallic bath; as it has been observed, such velocities are not reached for any of the cases. However, these are high enough to promote movement in the level of the free surface that gives rise to surface defects in the steel.

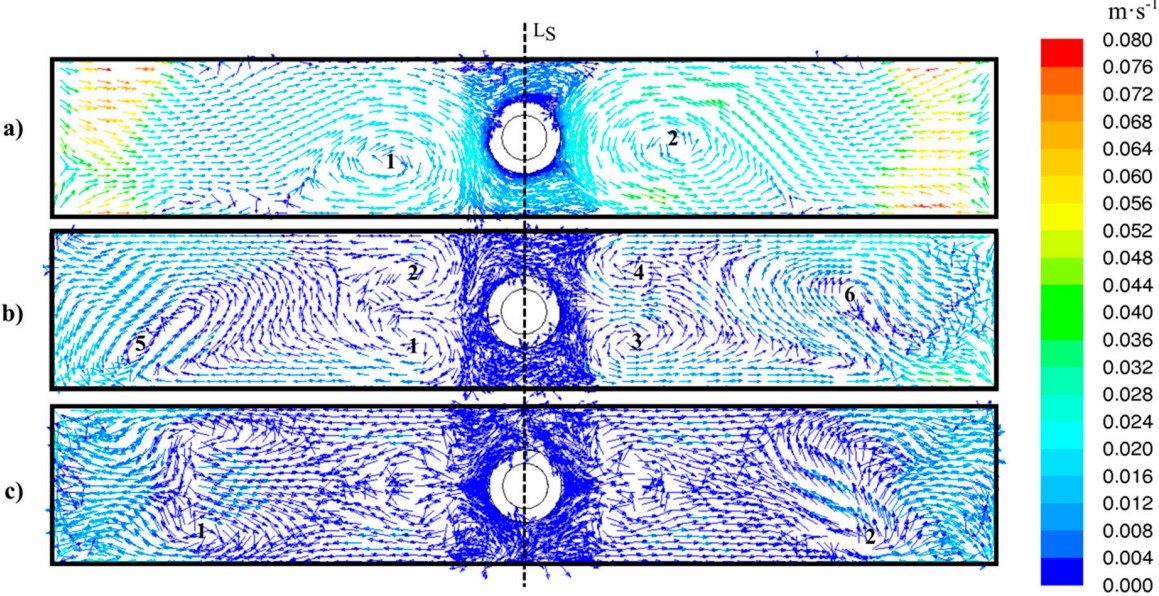

**Figure 6.** Velocity fields in the $P_V$ plane. (**a**) Case I, (**b**) Case II and (**c**) Case III.

### 3.3. Oscillations of Level

Figure 7 shows the level oscillations in $P_I$ at 410 s. On the left is a view of the upper area across the width of the mold, and on the right is the zoom indicated in the figure. In order to measure the maximum and minimum values of the oscillations, auxiliary lines with 5 mm separation between them were plotted taking the initial reference level that corresponds to the state of the fluid at rest. For all cases, a quasi-symmetric fluid dynamic behavior is observed in reference to the axis of symmetry of the SEN. Likewise, for the three cases the surface elevation in the region near the narrow wall is observed, since it is where the main upper recirculations are fed. However, the differences from one case to another are quite clear and vary in proportion to the fluid dynamic structure. In Figure 7a, that corresponds to Case I, it is observed that the oscillations pass over the initial reference level (0 mm) several times in wave form (higher order of Stokes) preferably. The positive heights of the crests reach values of up to 17 mm near the narrow walls of the mold and values in the valleys up to −10 mm with respect to the reference level. For Case II Figure 7b, compared to Case I, the decrease in waves is very relevant in most of the surface, with positive heights that do not exceed 5 mm near the narrow walls and values in the valleys around −13 mm in the area adjacent to the SEN. For Case III (Figure 7c), wave formation is not observed in this plane along the free surface, only a small perturbation adjacent

to the narrow wall of the mold with height less than 6 mm and a gradual descent of level almost imperceptible from the disturbance until the SEN. Finally, a depression is formed near the SEN with a value in the valley of less than −7 mm with respect to the reference level. The behavior for each case is reasonable according to the respective fluid dynamic structure. In addition, it is in accordance with the level of energy that is supplied through the main upper recirculations and the way in which it dissipates. However, as indicated [39] a static free surface can cause problems of overcooling and inhibition of steel-slag contact, causing some quality problems in the product.

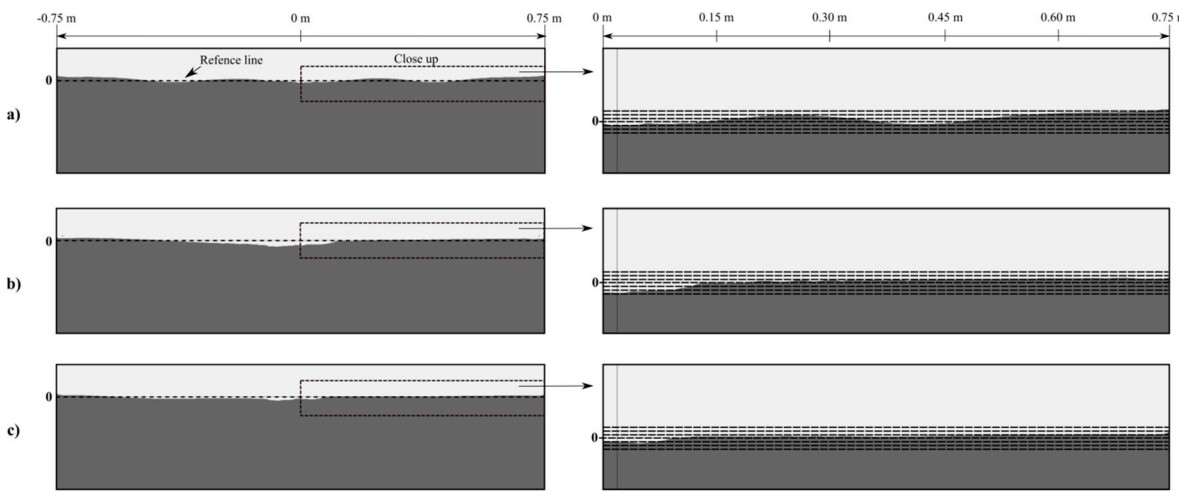

**Figure 7.** Meniscus level oscillations and close up. (**a**) Case I, (**b**) Case II, (**c**) Case III.

The level oscillations in a single plane in the mold have been reported [32]; however, being a three-dimensional and transient phenomenon, the oscillation profile changes significantly along such thickness. Figure 8 shows the level oscillations in $P_{II}$, $P_{III}$ and $P_{IV}$ at the simulation times of 640, 530 and 720 s for the three cases analyzed. In the three cases it is observed that in the areas close to the narrow walls, the level tends to reach its maximum levels, decreasing as the immersion depth of the SEN increases. On the contrary, the fluctuations of the level below the reference line tend to be located at the center of the mold, very close to the SEN. For Case I, it is seen that the level tends to oscillate with greater emphasis in $P_{II}$ and $P_{IV}$, showing oscillations of a higher order of Stokes, $P_{IV}$ being the plane on which the greatest elevations of the level are observed with respect to the reference line. For Case II, there is a noticeable reduction in the intensity of the oscillations in all planes analyzed, the level tends to fluctuate above the reference line almost over the entire width of the mold with the tendency to form small waves, mainly located near the SEN. While in Case III, a quasi-static free surface is seen without the appearance of waves as in the previous cases. In $P_{II}$ and $P_{IV}$, the level variations below the line are greater than in $P_{III}$, and it is even observed that these profiles are very similar to each other. It was observed that the behavior of the level oscillations along the thickness of the mold changes noticeably. In the $P_{III}$ plane the oscillation profile tends to be quasi-symmetric with respect to the axis of symmetry of the SEN.

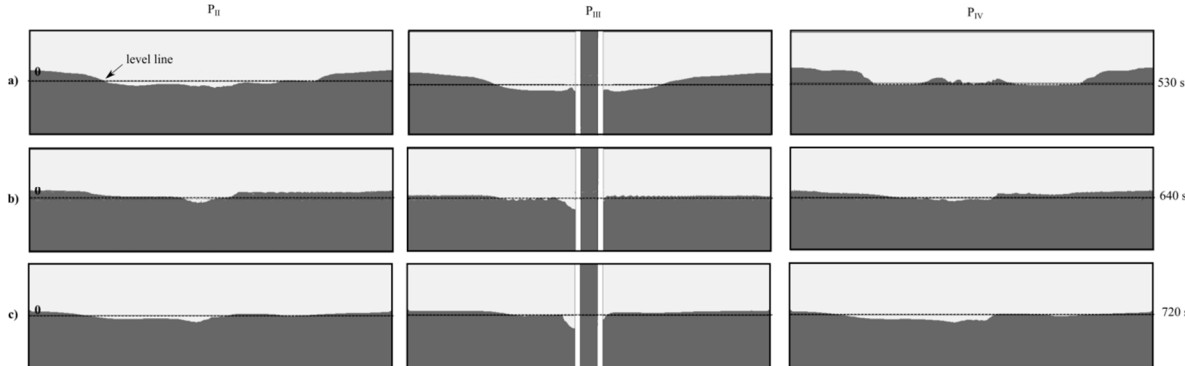

**Figure 8.** Meniscus level oscillations at different times and planes. (**a**) Case I, (**b**) Case II, (**c**) Case III.

### 3.4. Index F

The *F* index represents the force with which the jet hits the narrow walls of the mold, ascends towards the free surface and causes movement in said interface [31]. This parameter is defined by Equation (6) and is schematically represented in Figure 9.

$$F = \frac{\rho Q V_e (1 - \sin \theta_F)}{4D},$$

(6)

where: $Q$, is the voluminic flow entering the system $\text{m}^3 \cdot \text{s}^{-1}$; $\rho$, is the density of the fluid in $\text{kg} \cdot \text{m}^{-3}$; $V_e$, is the impact velocity of the discharge jet in $\text{m} \cdot \text{s}^{-1}$; $D$, is the distance from free surface to the impact point of the jet with the narrow wall in m; $\theta_F$, is the impact angle of the jet.

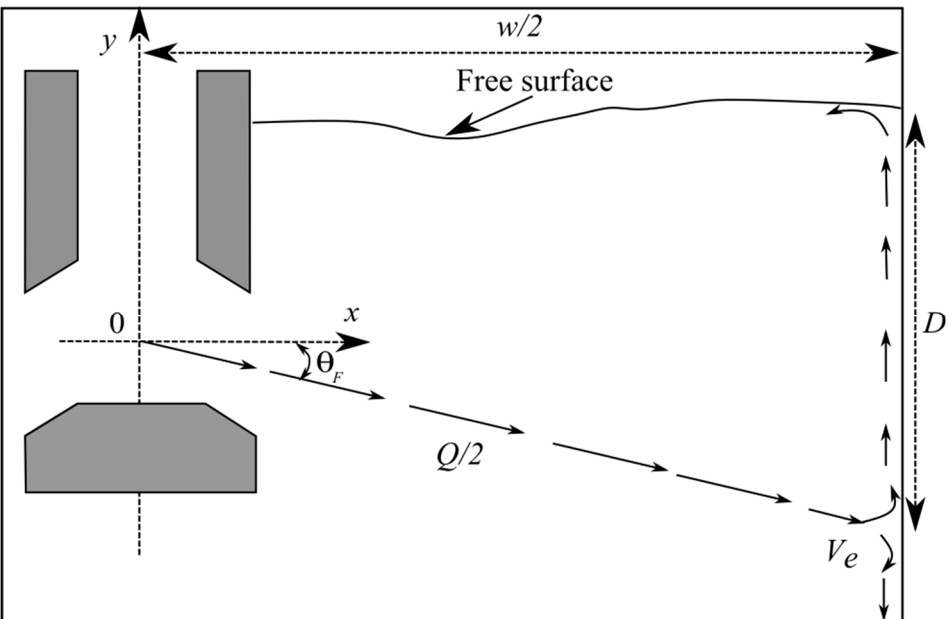

**Figure 9.** Scheme of *F* index.

Figure 10 shows the value of the *F* index with respect to the time for the left and right sides of the mold. It is observed that the behavior of this parameter is directly proportional to the depth of the SEN. For Case I, the maximum and minimum are found in the jet on the right side with times of ~680 and 530 s, respectively. In Case II, it can be noted that the maximum and minimum value are also presented on the right side at ~640 and 410 s, respectively. While for Case III, the maximum value is given on the right side at ~530 s and the minimum value at ~640 s on the left side. It should be mentioned that this parameter has an inverse behavior on each side of the mold, for example, at ~680 s for Case I, the right side has a maximum value and the left a minimum at a similar time, this happens for the three cases analyzed over time. If the value of the *F* index is related with the level oscillations shown in Figures 7 and 8, it can be seen that: when this index reaches its maximum value there is a greater elevation of the level of that side and a decrease of the level oscillations on the other side; i.e., the level oscillations increase linearly with the increase in the *F* value, similar results were reported by Teshima [31]. In addition, the *F* index decreases with increasing immersion depth of the SEN.

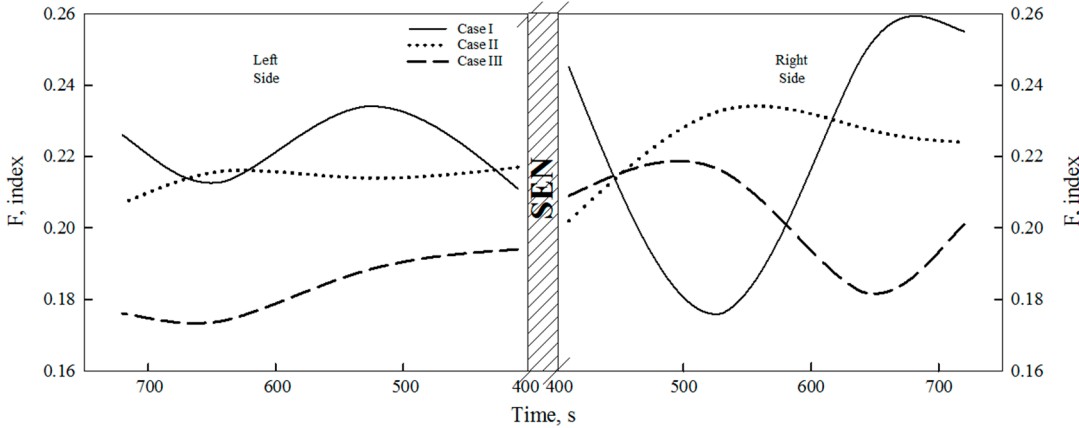

**Figure 10.** *F* index for all cases at different times in both sides of the mold.

### 3.5. Opening of the Oil Layer

Figure 11 shows the oil layer opening that simulates the slag for all cases. It is appreciated that the oil opens in the areas near the narrow walls (white zones). In addition, the thickness of the oil increases gradually from the opening to the walls of the SEN as can be seen in the isometric view. The thickening of the oil layer towards the SEN is due to the deformation stresses of the water/oil interface, which is caused by the higher speeds that ascend from the narrow walls. In relation to the opening percentage, Case I has 17.5%; Case II 11.5%; while Case III has 7.8% of the total area of the free surface. It is well known that a larger opening area could lead to the re-oxidation of the metal inside the mold, which is an undesirable phenomenon that must be avoided.

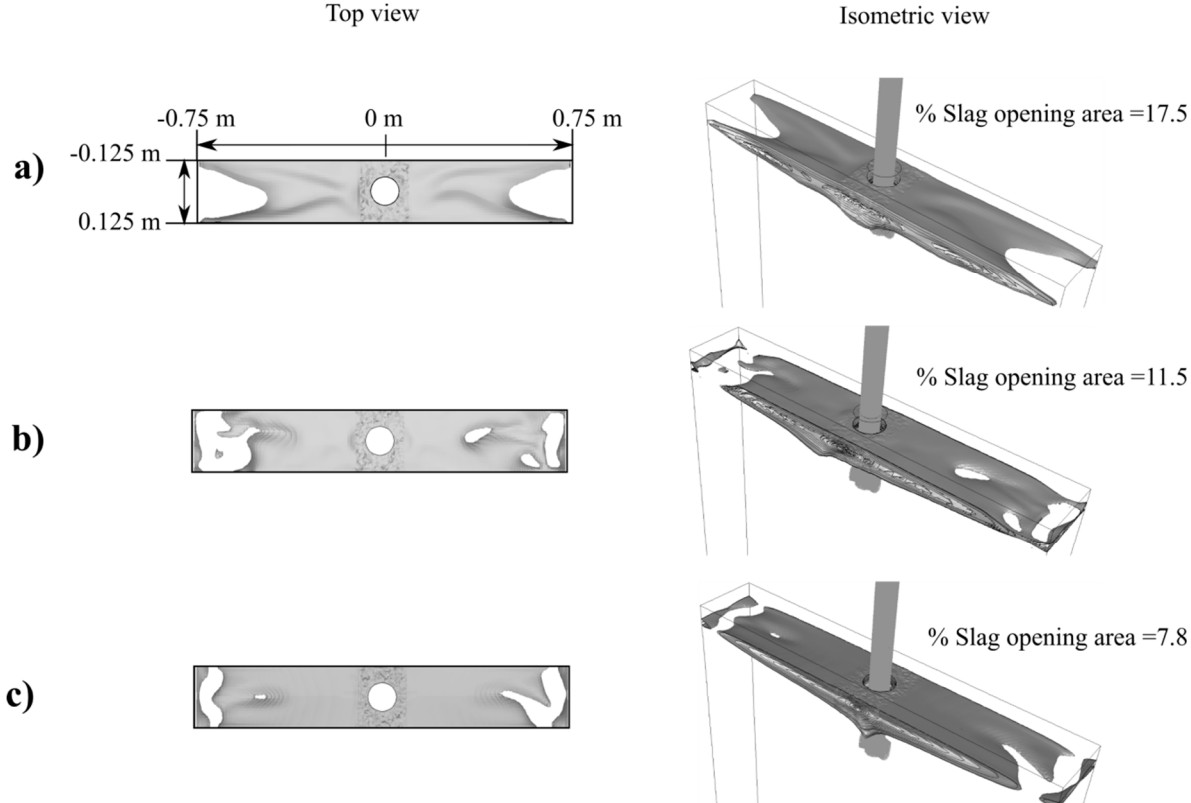

**Figure 11.** Oil layer opening. (**a**) Case I, (**b**) Case II, (**c**) Case III.

## 4. Conclusions

The multiphase mathematical simulation of the fluid dynamics was carried out for a continuous casting mold of conventional slabs through an analogue system water/air/vegetable oil, varying the immersion depth of the SEN. According to the results, the following is concluded:

1. A good agreement of the fluid-dynamic structure results obtained by the techniques of colorimetry, PIV and mathematical simulation was found.
2. As the immersion depth of the SEN increases, there is a decrease in the intensity and velocity of the flow directed towards the free surface, from ~0.08 m·s$^{-1}$ for Case I to 0.02 m·s$^{-1}$ for Case III.
3. Case I exhibits oscillations in the form of waves of second degree Stokes or higher, with large valleys and crests, with ranges of oscillation between +17 and −10 mm measured with respect to the reference level. On the other hand, elevations and valleys in the level were obtained in all the free surface of the mold with values of +5 to −13 mm, and of +6 to −7 mm in Cases II and III, respectively.
4. The opening of the oil layer was larger for Case I compared to Cases II and III, which can be related to the values of the index *F*, level of oscillations and fluid dynamic structure.

According to the analysis developed and the results obtained in this work, it is possible to recommend an operating range between 150 and 200 mm for the immersion depth of the SEN. The above is supported by the fact that the velocity fields around the SEN are of lower intensity, which implies less wear of the walls; the level oscillations are lower and similar in this range; and the oil layer remains more stable compared to the case of shallow depth.

**Author Contributions:** conceptualization, E.T.-A. and F.S.-S.; methodology, J.A.R.-B.; software, G.S.-D.; validation, F.S.-S., C.A.H.-B. and G.S.-D.; formal analysis, J.A.R.-B. and C.A.H.-B.; investigation, F.S.-S.; resources, G.S.-D.; data curation, F.S.-S.; writing—original draft preparation, C.A.H.-B. and J.A.R.-B.; writing—review and editing,

J.A.R.-B. and C.A.H.-B.; visualization, E.T.-A.; supervision, G.S.-D.; project administration, E.T.-A.; funding acquisition, G.S.-D.

**Funding:** This research received no external funding.

**Acknowledgments:** The authors want to acknowledge to the UMSNH-FIM, TecNM-ITM, CÁTEDRAS-CONACyT, and CONACyT for their continuous support to Doctoral Scholarship 294877 and SNI for the permanent support to the academic groups of Modeling of Metallurgical Processes and Thermofluids.

**Conflicts of Interest:** The authors declare no conflict of interest.

## Appendix A

**Table A1.** Nomenclature and Greek letters description.

| Nomenclature | |
|---|---|
| $A_{inlet}$ | Inlet Area |
| $A_{outlet}$ | Outlet area |
| $D$ | Distance from free surface to the jet impact point |
| $F$ | *F* index |
| $g$ | Gravitational acceleration |
| $L_s$ | Symmetry line |
| $n$ | Number of phases |
| $p$ | Pressure |
| $P_I, P_{II}, P_{III}, P_{iv}$ y $P_v$ | Planes of analysis |
| $Q$ | Volumetric flow rate of fluid |
| $S_{user}$ | User-defined mass source |
| $t$ | Time |
| $u$ | Velocity of the fluid |
| $u'_i, u'_j, u'_k$ | Instantaneous velocity vector |
| $V_e$ | Jet impact velocity |
| $V_{inlet}$ | Inlet velocity |
| $V_{casting}$ | Casting velocity |
| $w$ | Mold width |
| $x_i, x_j, x_k$ | Spatial vector components |
| **Greek Symbols** | |
| $\alpha$ | Mass fraction |
| $\beta$ | Coefficient of thermal volumetric expansion |
| $\varepsilon_{ikm}$ | Levi-Civita tensor |
| $\zeta_{kj}, \zeta_{ik}$ | Kronecker delta |
| $\theta = f(\mu_t, Pr_t, T)$ | $\mu_t$: turbulent viscosity, $Pr_t$: turbulent Prandtl, $T$: temperature) |
| $\theta_F$ | Jet impact angle |
| $\mu$ | Viscosity of the fluid |
| $\rho$ | Density |
| $\sigma$ | Surface tension |
| $\Omega_k$ | Rotation vector |

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
