# Peer review of "Analysis of the Depth of Immersion of the Submerged Entry Nozzle on the Oscillations of the Meniscus in a Continuous Casting Mold"

_metals, doi:10.3390/met9050596_

Round 1

Reviewer 1 Report

Please provide higher resolution for the vector fields in Fig. 4-6. It is not possible to see sufficient details while zooming.

Caption of the Fig. 7 is not in English!

Author Response

Please provide higher resolution for the vector fields in Fig. 4-6. It is not possible to see sufficient details while zooming.

R: All images were change from 300 to 600 dpi to improve the visualization while zooming.

Caption of the Fig. 7 is not in English!

R: Figure caption was changed to English language as indicated in page 10, line 261.

Reviewer 2 Report

The paper “Analysis of the Depth of Immersion of the Submerged Entry Nozzles on the Oscillations of the Level of a Continuous Casting Mold” depicts the fluid flow of the molten steel in the primary cooling zone. Experimental work, including development of the mould water model with the SEN and the PIV measurements, along with the numerical modelling has been presented by the Authors. For sure, in the continuous casting field, there is a space for research related to the positioning of the SEN. After reading the content of the presented paper following questions arose:

1. I do understand a trend, which was established in a numerical modeling of the molten steel fluid flow, but what is the point of those type of simulation? I mean a fluid flow simulation without thermal calculation? Do the Authors know how the fluid flow would be distorted by the presence of the solid shell in the mould?

2.  What was the thickness of the oil layer (used in the water model)?

3. Did the Authors use commercial software Ansys Fluent in order to perform numerical calculation? Such information should be added in the text.

4. All symbols presented in the formulas have to be explained.

5. Line 46-54: Presented comments are true for typical steel production e.g. low carbon steel. For more demanding grades of steel other casting technology has to be implemented.

6. At the end, the Authors recommend the immersion depth for SEN between 150 - 200 mm. From my point of view, such conclusion, without control of the solid shell thickness leaving the mould and a fixed length of the mould isn't complete and proper. What is a technological aspect of the presented research?

Author Response

1.     fluid flow, but what is the point of those type of simulation? I mean a fluid flow simulation without thermal calculation? Do the Authors know how the fluid flow would be distorted by the presence of the solid shell in the mould?

R: The point is to identify the immersion depth of the SEN on the surface disturbances, which are related to the F index, which was designed as an industrial parameter to determine the quality of the steel. On the other hand, the corresponding author published a related work which analyzed the influence of including the solidification model on fluid dynamics in the mold, where it was found that including the solidification model does not significantly affect the fluid dynamics structure inside the mold.

(https://www.researchgate.net/publication/257872873_SHELL_GROWTH_DYNAMICS_IN_A_CURVED_SLAB_MOLD_AFFECTED_BY_FLUID_FLOW_HEAT_TRANSFER_AND_FLUX_INFILTRATION)

Therefore, in the present work, the solidification model was not solved, which also allowed to decrease the computational cost 

2.     What was the thickness of the oil layer (used in the water model)?

R: In section 2.3 Boundary conditions, it was mentioned the oil layer thickness on page 4 line 121. Furthermore, the next sentence was added in section 3.1 Validation of the Model. On page 6, lines 153 to 154: “For the experimental validation, the same conditions of the mathematical model were used”

3.     Did the Authors use commercial software Ansys Fluent in order to perform numerical calculation? Such information should be added in the text.

R: Yes, it was already indicated on page 5 line 130.

4.     All symbols presented in the formulas have to be explained.

R: The authors considered it convenient to include the symbols and Greek letters list at the end of the manuscript.

5.     Line 46-54: Presented comments are true for typical steel production e.g. low carbon steel. For more demanding grades of steel other casting technology has to be implemented.

R: The authors agree with the comment, about the implementation of new technologies for other grades of steel; that is, the coverage powders must be analyzed, as well as the chemistry of the refractory coating. The above with the purpose of reducing erosion due to chemical wear. However, the turbulence caused by the movement of the fluid will always be an erosion factor when working at low depth of immersion of the SEN.

6.     At the end, the Authors recommend the immersion depth for SEN between 150 - 200 mm. From my point of view, such conclusion, without control of the solid shell thickness leaving the mould and a fixed length of the mould isn't complete and proper. What is a technological aspect of the presented research?

R: As mentioned above, there is no significant effect in including the solidification model on the fluid dynamics structure. The authors agree that including the energy equation and the solidification model coupled with fluid dynamics would make the proposed study more realistic. On the other hand, it is important to mention that the authors are working on determining correlations to estimate the heat fluxes in the mold walls experimentally. On the other hand, in this work, two meters of length were simulated, in order to minimize the effects of boundary conditions at the exit. And in the physical model baffles were installed to guarantee the casting speed without suction effects, these deflectors are visible in the narrow face of the mold in the lower part of Figure 3. The approach that has been given to the work was to analyze the effects on the oscillation due to the immersion depths of the SEN.